# Resveratrol and Vascular Function

**DOI:** 10.3390/ijms20092155

**Published:** 2019-04-30

**Authors:** Huige Li, Ning Xia, Solveig Hasselwander, Andreas Daiber

**Affiliations:** 1Department of Pharmacology, Johannes Gutenberg University Medical Center, 55131 Mainz, Germany; xianing@uni-mainz.de (N.X.); sohassel@uni-mainz.de (S.H.); 2Center for Cardiology, Cardiology I - Laboratory of Molecular Cardiology, Johannes Gutenberg University Medical Center, 55131 Mainz, Germany

**Keywords:** resveratrol, endothelium, endothelial nitic oxide synthase, sirtuin 1, cardiovascular disease, vascular function

## Abstract

Resveratrol increases the production of nitric oxide (NO) in endothelial cells by upregulating the expression of endothelial NO synthase (eNOS), stimulating eNOS enzymatic activity, and preventing eNOS uncoupling. At the same time, resveratrol inhibits the synthesis of endothelin-1 and reduces oxidative stress in both endothelial cells and smooth muscle cells. Pathological stimuli-induced smooth muscle cell proliferation, vascular remodeling, and arterial stiffness can be ameliorated by resveratrol as well. In addition, resveratrol also modulates immune cell function, inhibition of immune cell infiltration into the vascular wall, and improves the function of perivascular adipose tissue. All these mechanisms contribute to the protective effects of resveratrol on vascular function and blood pressure in vivo. Sirtuin 1, AMP-activated protein kinase, and estrogen receptors represent the major molecules mediating the vascular effects of resveratrol.

## 1. Resveratrol and its Molecular Targets

The polyphenolic phytoalexin 3,5,4′-trihydroxy-trans-stilbene, better known under its trivial name resveratrol, can be found in numerous plants, such as white hellebore (*Veratrum grandiflorum*), mulberry (*Morus rubra*), peanut (*Archis hypogaea*), and grapes (*Vitis vinifera*) [1,2,3].

Since resveratrol gained popularity in 1992 [4], many targets responsible for its pharmacological effects have been identified [5,6]. These targets can be divided into those which directly interact with resveratrol (over 20) and those whose effects are indirectly changed, e.g., by modulation of their expression level [6]. Regarding vascular function, especially the estrogen receptor (ER), the NAD^+^-dependent, class III histone deacetylase sirtuin 1 (SIRT1), the nuclear factor-erythroid-derived 2-related factor-2 (Nrf2), and the AMP-activated protein kinase (AMPK) are of particular importance [7].

Resveratrol can directly activate SIRT1 on certain substrates [8,9]. Indirectly, resveratrol increases SIRT1 activity either through elevation of intracellular NAD^+^ concentration, which is dependent on an inhibition of phosphodiesterase (PDE) [10,11], or through an enhancement of the binding of SIRT1 to lamin A, an endogenous SIRT1 activator [12]. In addition, the SIRT1-dependent effects of resveratrol in vivo are partially attributable to an upregulation of SIRT1 expression by the compound [13,14,15].

Nrf2 is an indirect target protein of resveratrol. For its stimulation, concentrations of resveratrol lower than 1 µM are sufficient (lower than that for SIRT1) [16], making its activation possible by using resveratrol as a dietary supplement [2]. Nrf2 interacts with antioxidant-response elements after its translocation to the nucleus. Here, it triggers the expression of phase II and antioxidant defense enzymes, like heme oxygenase-1 (HO-1) [16].

As mentioned before, resveratrol has been linked to PDE inhibition. One result of this effect is the phosphorylation of AMPK [11]. Another suggested pathway for AMPK activation is based on LKB1 stimulation, e.g., by reduction of the intracellular ATP level [17,18] or its deacetylation by SIRT1 [19,20]. Interestingly, AMPK and SIRT1 seem to function synergistically and bolster one another [21,22].

Resveratrol can directly bind the estrogen receptors (ERs) [23]. Very low (nanomolar) concentrations of resveratrol are enough for the ER-mediated stimulation of endothelial NO synthase (eNOS) [24,25]. Additionally, resveratrol-mediated ER-stimulation has also been linked to HO-1 upregulation as well as to Nox downregulation [26].

Also, the effects of resveratrol in vivo may also involve its actions on potassium channels [27,28,29], gut microbiota [30,31], and circadian gene expression [32].

## 2. Effects of Resveratrol on Endothelial Cells

The endothelium, consisting of a single layer of flat, longish endothelial cells, covers the inner walls of blood vessels. Vascular endothelial cells are unique in their property of holding Weibel-palade bodies, the depot for the Von Willebrand factor which is crucial for hemostasis maintenance. Apart from its function in maintaining blood coagulation and serving as starting point for angiogenesis, the endothelium also provides a semi-permeable barrier to regulate the transfer of electrolytes, macromolecules, and fluid between the intravascular and the extravascular space. Endothelial cells synthesize important vasoactive substances, including prostacyclin, NO, and the vasocontractile endothelin-1 (ET-1). Therefore, endothelial cells are key regulators of blood pressure and vascular tone [33].

### 2.1. Resveratrol Enhances Endothelial NO Production

Under physiological conditions, the endothelial NO synthase (eNOS) is the main producer of vascular NO [34]. It confers antithrombotic, antihypertensive, and anti-atherosclerotic effects [34,35]. Endothelial NO reaches the smooth muscle cells (SMC) by diffusion and causes vasodilation [36]. In the blood, eNOS-produced NO prevents platelet aggregation and adhesion. The anti-atherosclerotic properties of NO include the prevention of leukocyte adhesion to and migration into the vascular wall as well as the repression of low-density lipoprotein oxidation and the proliferation of vascular smooth muscle cells [34,37,38,39]. Suppression of the eNOS gene in mice leads to blood pressure elevation [40] and atherosclerosis aggravation [41,42]. Recently, it has been shown that eNOS-produced NO might be involved in mitochondrial biogenesis boosting [13] and could be partly responsible for the observed antiaging effects in calorie restriction studies [43]. In addition, it has been demonstrated that eNOS-knockout mice exhibit insulin resistance as well as hyperinsulinemia [44], while overexpression of eNOS protects mice fed with a high-fat diet (HFD) from pathological weight gain [45].

Resveratrol increases endothelial NO production through multiple mechanisms (Figure 1), including upregulation of eNOS expression, enhancement of eNOS enzymatic activity, and prevention of eNOS uncoupling [46].

#### 2.1.1. Resveratrol Upregulates eNOS Expression

Resveratrol increases eNOS expression in endothelial cells. eNOS is constitutively expressed in the endothelium; its expression level changes in response to various stimuli and compounds [35,47]. We have previously shown that resveratrol itself [48] as well as resveratrol-containing red wines [49,50] enhance eNOS expression in endothelial cells (Table 1). Interestingly, this phenomenon is not dependent on ERs but associates with transcriptional and posttranscriptional mechanisms [48]. In endothelial cells, knock-down of the SIRT1 gene via siRNA inhibits the up-regulation of eNOS by resveratrol treatment [13]. Correspondingly, an endothelium-specific overexpression of SIRT1 leads to elevated eNOS expression [51]. Therefore, resveratrol-induced up-regulation of eNOS is likely to be SIRT1-dependent. Strengthening this suggestion, we were recently able to show the involvement of FOXO factors as downstream targets of SIRT1 in resveratrol effects [14].

#### 2.1.2. Resveratrol Increases eNOS Activity

In addition to eNOS expressional upregulation, resveratrol also enhances enzymatic activity of eNOS through post-translational modifications. Phosphorylation of eNOS at Ser-1177 is an activating modification [52,53]. Resveratrol has been shown to increase eNOS Ser-1177 phosphorylation in cultured endothelial cells [24,25]. For this effect, resveratrol concentrations of the nanomolar range are sufficient. The underlying signaling pathway includes the estrogen receptor ERα, G-protein Gα, caveolin-1 (Cav-1), the tyrosine kinase c-Src, and the MAP kinase Erk1/2 [24,25]. In concentrations of the micromolar range, phosphorylation of eNOS at the Ser-1177 by resveratrol also involves the activation of the AMPK [54,55]. Additionally, resveratrol enhances eNOS activity by inducing SIRT1-mediated deacetylation of eNOS at Lys-496 and Lys-506 [56,57].

An established endogenous eNOS inhibitor is asymmetric dimethylarginine (ADMA) [58]. Its degrading enzyme dimethylarginine dimethylaminohydrolase (DDAH) was found to be down-regulated by high concentrations of glucose, resulting in intracellular ADMA accumulation. This development, however, can be prevented by pre-treating of the cells with resveratrol [59].

Another negative regulator of eNOS is Cav-1 [60]. It was shown in vitro (endothelial cells [25,61]) and in vivo (rat heart [62,63]) that resveratrol targets these protein–protein interactions both by reduction of the Cav-1 expression level and by restriction of the Cav-1/eNOS association [62,63].

#### 2.1.3. Resveratrol Prevents eNOS Uncoupling

Importantly, resveratrol also prevents eNOS uncoupling. This terminology describes the phenomenon of superoxide production by eNOS under pathological conditions [64,65,66]. Major mechanisms underlying eNOS uncoupling are the lack of the eNOS substrate l-arginine or the eNOS cofactor BH4 as well as eNOS S-glutathionylation [38,64,66]. Causes for BH4 insufficiency are its oxidation by potent oxidizing agents, such as peroxynitrite and superoxide [67,68,69,70]. This event and the following eNOS uncoupling though, can be prevented by resveratrol due to its antioxidant effects. Additionally, resveratrol boosts BH4 biosynthesis by enhancing the expression of GTP cyclohydrolase 1, an essential enzyme for BH4 production [71]. We previously described the vicious circle resulting from eNOS uncoupling leading to a potentiation of oxidative stress as a central mechanism adding to cardiovascular diseases [64,65,66]. Resveratrol intervenes in this circle by avoiding eNOS uncoupling and excess oxidative stress (Table 2).

### 2.2. Resveratrol Reduces Endothelial Oxidative Stress

Since the capacity of resveratrol to directly eliminate free radicals is lower than that of already established antioxidant reagents, the means by which it diminishes oxidative stress in vivo are more likely due to its ability to regulate gene expression (Figure 2) [15]. Indeed, resveratrol is a potent gene regulator [76]. At a concentration of 0.1 µM, resveratrol has been shown to upregulate 127 genes and downregulate 233 genes in cultured human umbilical vein endothelial cells (HUVEC) [33].

In cultured endothelial cells, resveratrol enhances the expression of phase 2 enzymes and antioxidant enzymes, including NAD(P)H:quinone–oxidoreductase 1 (NQO1), γ-glutamylcysteine synthetase (GCLC), HO-1, SOD1, SOD2, glutathione peroxidase 1 (GPx1), and catalase [16,71,73,74]. The expression [71,73] and activity [72] of NADPH oxidases, major ROS-producing enzymes in endothelial cells, are reduced by resveratrol (Table 2). Furthermore, resveratrol improves mitochondrial biogenesis and thus diminishes mitochondrial superoxide production [75].

SIRT1 and Nrf2 play important roles in the expressional regulation of redox genes by resveratrol [15]. Resveratrol-induced upregulation of SOD1, SOD2, GPx1, and catalase requires SIRT1, whereas Nrf2 is responsible for the upregulation of HO-1, NQO1, and GCLC by resveratrol [16,71,73,74,75].

### 2.3. Resveratrol Reduces Endothelin-1 Synthesis

ET-1 is a highly potent vasoconstrictor [33]. Overproduction of ET-1 is implicated in the development of vascular disease and atherosclerosis [77]. Interestingly, resveratrol has been shown to reduces ET-1 synthesis in endothelial cells and smooth muscle cells (Table 3), as well as in the plasma of resveratrol-treated rabbits [78]. It is conceivable that both the enhanced production of NO and reduced synthesis of ET-1 are involved in the vasodilation and blood pressure reduction by resveratrol in vivo.

## 3. Effects of Resveratrol on Vascular Smooth Muscle Cells

### 3.1. Resveratrol Reduces Oxidative Stress in Smooth Muscle Cells

Resveratrol has been shown to produce biphasic effects on HO-1 expression in human aortic SMC in a NF-κB-dependent manner, with a HO-1 induction at low concentrations of resveratrol (1–10 μM) and a reduction at high concentrations of resveratrol (≥20 μM) [83]. This phenomenon has not been observed in later studies. In in rat aortic SMC, resveratrol (1–30 µM) causes a concentration-dependent upregulation of HO-1 mediated by Nrf2 [84].

Treatment of cultured rat aortic SMC with resveratrol (25–100 µM) leads to upregulation of SOD, catalase, glutathione, glutathione reductase, glutathione peroxidase, glutathione S-transferase (GST) and NOQ1 [85]. GST and NQO1 are typical phase 2 enzymes responsible for detoxification of both ROS and electrophilic xenobiotics [85]. The induction of such cellular antioxidants and phase 2 enzymes by resveratrol protects the cells from oxidative stress and oxidants-induced cytotoxicity (Table 4).

### 3.2. Resveratrol Inhibits Smooth Muscle Cell Proliferation

Normally differentiated vascular smooth muscle cells (VSMC) express a specific set of contractile proteins, have low synthetic activity, and proliferate slowly [86]. Upon stimulation by growth factor or in response to vascular injury, VSMC can change their phenotype through de-differentiation, proliferation, and migration to the site of injury [86]. While the phenotypic plasticity in VSMC is essential for the maintenance and repair of the vasculature, excessive proliferation of VSMC promotes the development of atherosclerosis, restenosis, and pulmonary hypertension [86,87].

Resveratrol has been shown to inhibit VSMC proliferation induced by various mitogens (Table 4). Depending on the mitogenic stimuli, the molecular mechanisms include inhibition of PI3K/Akt/mTOR pathway or cell cycle arrest (Table 4). In addition, antiproliferative effects in VSMC have also been reported for the resveratrol tetramer vitisin B [88] and for pterostilbene, a natural dimethylated analog of resveratrol [89]

Oral administration of resveratrol has been shown to significantly suppress intimal hyperplasia in mouse models of wire-injured arteries [84,90]. Intraperitoneal injection of resveratrol also inhibited the development of intimal hyperplasia in the rat carotid artery injury model [91].

Because of the low bioavailability of systemically administrated resveratrol, local delivery may represent a promising method improving the in vivo efficacy of the compound. Resveratrol coated on a stent reduced stenosis in rat carotid artery by 65% [92]. In a recent study of rat carotid balloon angioplasty, resveratrol was applied in Pluronic gel around the injured artery [93]. Strikingly, periadventitial application of resveratrol significantly improved all three major pathologies contributing to restenosis: VSMC hyperplasia in the intima, impairment of re-endothelialization, and constrictive remodeling [93]. Periadventitial delivery of resveratrol produced a much greater neointima-inhibiting effect (86%) than systemic resveratrol administration reported in previous studies. Moreover, post-surgery endothelial recovery was accelerated by resveratrol without causing constrictive remodeling [93]. These are compelling advantages over drug-eluting stents currently used in clinical settings.

### 3.3. Resveratrol Prevents Arterial Stiffness and Vascular Remodeling

In addition to inhibition of VSMC proliferation, resveratrol also prevents vascular remodeling and arterial stiffness. In cultured rat aortic SMC stimulated with advanced glycation end-products (AGEs), resveratrol reduces TGF-β1 expression and collagen synthesis [95]. In human VSMC, TNF-α-induced expression of matrix metalloproteinase (MMP)-9 is inhibited by resveratrol [100].

Resveratrol has been shown to prevent high-fat, high-sucrose diet-induced arterial stiffness in mice [104]. This effect is likely to be mediated by SIRT1, because similar effects can be achieved with SIRT1 activators or by global SIRT1 overexpression [104]. Interestingly, overnight fasting acutely decreased arterial stiffness in wildtype mice but not in mice lacking SIRT1 in VSMC. Conversely, VSMC-specific SIRT1 overexpression prevented diet-induced arterial stiffness. The antistiffness effect of SIRT1 has been attributable to its anti-inflammatory and antioxidant properties mediated by NF-κB inhibition and downregulation of VCAM-1 and p47phox [104].

Resveratrol downregulates angiotensin II (AngII) type 1 receptor expression in VSMC through SIRT1 activation [105]. Moreover, resveratrol treatment decreases serum AngII level and the aortic expression of prorenin receptor (PRR) and angiotensin converting enzyme (ACE) and increases serum Ang(1–7) level and the expression of ACE2, AngII type 2 receptor (AT2R), and Mas receptor (MasR). These mechanisms (modulation of the renin–angiotensin system) have been made responsible for the protective effects of resveratrol on aging-induced vascular fibrosis [106] and kidney fibrosis [107].

## 4. Effects of Resveratrol on Immune Cells

Although leukocytes do not belong to the resident cells of the vascular wall, they contribute to the pathogenesis of cardiovascular disease [38]. Activated endothelial cells express adhesion molecules on their surface, leading to cell–cell interaction and immune cell infiltration into the vascular wall [108]. Resveratrol modulates the function of immune cells, including CD4^+^/CD8^+^ T cells [109], regulatory T cells [110], natural killer (NK) cells [111], polymorphonuclear leukocytes [109], and monocytes [110,112].

Resveratrol prevents the interaction between immune cells and endothelial cells. On the one hand, resveratrol (10–100 µM) inhibits the expression and binding activity of chemokine receptors on leukocytes, e.g., CCR2 on monocytes [110]. On the other hand, resveratrol (0.1–1 µM) also reduces the expression of adhesion molecules on endothelial cells [113]. Reduced macrophage infiltration and ameliorated vascular inflammation by resveratrol have also been demonstrated in vivo [114].

## 5. Effects of Resveratrol on PVAT

The perivascular adipose tissue (PVAT) is now considered an important regulator of vascular function by secreting a large number of vasoactive substances, including NO [115,116]. We have recently seen that the importance of PVAT-eNOS may even exceed that of endothelial eNOS under certain conditions [117]. In the thoracic aorta of diet-induced obese mice, for example, vascular dysfunction can only be observed if PVAT is kept in place. In aortae without PVAT of mice with obesity, vascular function remains normal [117,118]. This observation has led to the hypothesis that the reduction of NO-dependent vasodilation in obese mice is not due to a dysfunction of eNOS in the endothelium but in the PVAT [117]. Indeed, vascular function of the obese mice can be normalized by pharmacological improvements of PVAT-eNOS function [117,118].

Interestingly, resveratrol has been shown to improve PVAT function [119,120]. The acetylcholine-induced relaxation of aortae from normal rats is inhibited by conditioned media derived from PVAT of rats fed with fructose [119] or HFD [120]. The fructose- and HFD-induced PVAT dysfunction is largely reversed by an oral treatment with resveratrol (20 mg/kg for 8 weeks) [119,120]. Similar PVAT dysfunction can be induced by an in vitro incubation of PVAT with palmitic acid. The palmitic acid-induced PVAT dysfunction is normalized by resveratrol (10 µM), an effect that is prevented by either the AMPK inhibitor compound C or the SIRT1 inhibitor nicotinamide [119]. Because these experiments are performed using PVAT-derived conditioned media, the observed protective effects of resveratrol can be clearly attributed to PVAT.

## 6. Effects of Resveratrol on Vascular Function and Blood Pressure in Vivo

Antihypertensive effects of resveratrol have been observed in several animal models [7,121,122]. These include genetic hypertension (e.g., spontaneously hypertensive rats [SHR]) [123,124], AngII-induced hypertension [124], renal models of hypertension [125,126], DOCA salt hypertension [127], programmed hypertension [128] and diet-induced metabolic syndromes [129,130,131]. Increased endothelial NO production, reduced vascular oxidative stress, and the resulting improvement of vascular function are likely the major molecular mechanisms underlying the blood pressure-lowering effects of resveratrol (Table 5).

Interestingly, fecal microbiome transplants from resveratrol-treated healthy mice reduce the systolic blood pressure and left ventricular mass of hypertensive mice [30].

## 7. Conclusions

Resveratrol increases endothelial NO production, decreases ET-1 synthesis, reduces vascular oxidative stress, and prevents smooth muscle proliferation, vascular remodeling, and arterial stiffness. In addition, resveratrol also inhibits immune cells infiltration into the vascular wall and mitigates vascular inflammation. All these mechanisms contribute to the in vivo effects of resveratrol on vascular function and blood pressure.

## Figures and Tables

**Figure 1 ijms-20-02155-f001:**
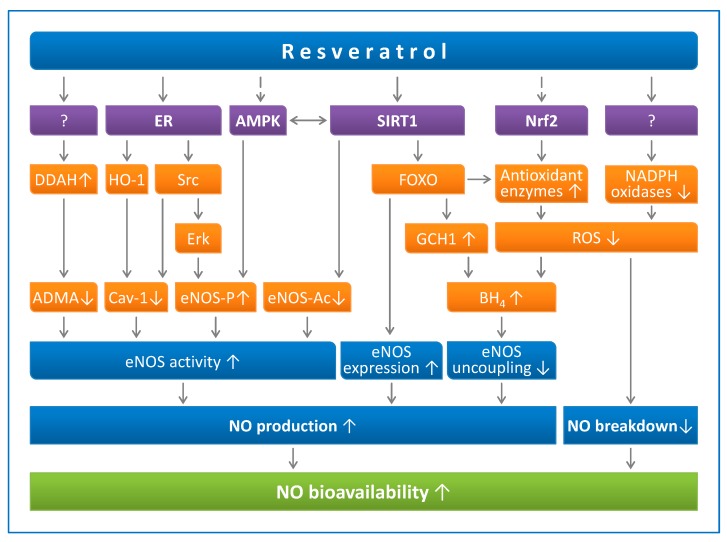
Resveratrol enhances NO production and prevents NO breakdown. Resveratrol can activate sirtuin 1 (SIRT1) directly (in a substrate-dependent manner) or indirectly (by either inhibiting phosphodiesterases or enhancing the effect of lamin A). SIRT1 stimulates endothelial NO synthase (eNOS) activity through deacetylation, enhances eNOS expression by deacetylating Forkhead box O (FOXO) transcription factors, and prevents eNOS uncoupling by upregulating GTP cyclohydrolase 1 (GCH1), the rate-limiting enzyme in tetrahydrobiopterin (BH4) biosynthesis. AMP-activated protein kinase (AMPK) and nuclear factor-erythroid-derived 2-related factor-2 (Nrf2) are indirect targets of resveratrol. AMPK phosphorylates eNOS at serine 1177. eNOS can also be phosphorylated by Erk1/2, which is stimulated by a pathway involving estrogen receptors (ER) and the tyrosine kinase Src. Caveolin-1 (Cav-1) is an eNOS-interacting protein that negatively regulates eNOS activity. Asymmetric dimethylarginine (ADMA) is an endogenous eNOS inhibitor that is degraded by dimethylarginine dimethylaminohydrolase (DDAH). The resveratrol targets for DDAH upregulation or for NADPH oxidase downregulation have not been identified so far. Reproduced from Xia et al. *Molecules*. **2014** [46], under the terms of the Creative Commons Attribution-Noncommercial License CC BY-NC.

**Figure 2 ijms-20-02155-f002:**
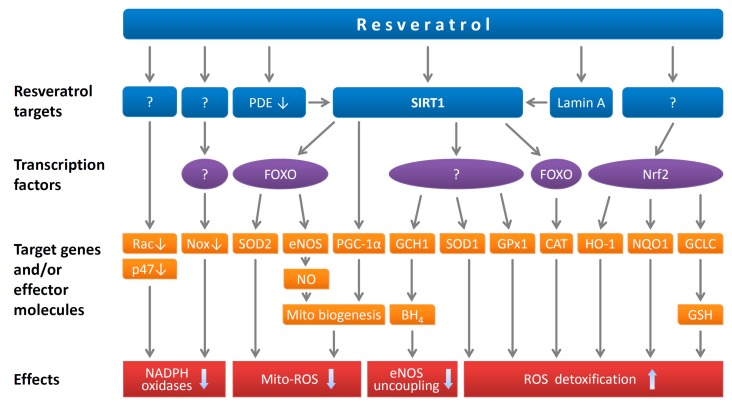
Antioxidant effects of resveratrol. Resveratrol inhibits NADPH oxidase-mediated reactive oxygen species (ROS) production by downregulation of the catalytic subunits (NOX proteins) and by inhibiting membrane translocation of Rac1 and inhibiting phosphorylation of p47phox. Resveratrol directly activates SIRT1 on certain substrates. It can also activate SIRT1 indirectly by potentiating the activation effect of lamin A or via a pathway involving phosphodiesterase (PDE) inhibition that leads to elevation of cellular NAD^+^. Among the established SIRT1 targets, FOXO transcription factors contribute to the antioxidant effects of resveratrol by upregulating antioxidant enzymes (e.g., SOD1, SOD2, GPx1 and catalase, CAT) and eNOS. SIRT1 inhibits mitochondrial superoxide production by stimulating mitochondrial biogenesis, which is mediated by PGC-1α deacetylation and by NO-dependent mechanisms. The upregulation of GCH1 leads to enhancement of BH4 biosynthesis and prevention of eNOS uncoupling. In addition, resveratrol upregulates a number of antioxidant enzymes by activating Nrf2. Reproduced from Xia et al. *Br. J. Pharmacol.*
**2017** [15] with permission. Copyright © 2017 John Wiley and Sons (Hoboken, NJ, USA).

**Table 1 ijms-20-02155-t001:** Resveratrol increases NO production in endothelial cells.

Cell Type	Effective Concentration	Effects	Reference
HUVEC	10–100 µM	eNOS↑; NO↑	[48]
EA.hy 926	10–100 µM	eNOS (via SIRT1/FOXO)↑; NO↑	[14,48]
HCAEC	1–100 µM	eNOS (via SIRT1)↑	[13]
HUVEC	0.1 µM	eNOS↑; VEGF↑; ET-1↓	[33]
BAEC, HUVEC	1–100 nM	p-eNOS↑ (via ERα & Erk1/2)	[24]
HUVEC	1–100 µM	p-eNOS↑ (via AMPK)	[54]
STA	50 µM	p-eNOS↑ (via AMPK)	[55]
RAEC	100 µM	Ac-eNOS↓	[56]

Ac-eNOS; acetylated eNOS; BACE, bovine aortic endothelial cells; HCAEC, human coronary arterial endothelial cells; HUVEC, human umbilical vein endothelial cells; p-eNOS, phosphorylation of eNOS at serine 1177; STA, superior thyroid artery; RAEC, rat aortic endothelial cells; VEGF, vascular endothelial growth factor.

**Table 2 ijms-20-02155-t002:** Resveratrol reduces oxidative stress in endothelial cells.

Cell Type	Effective Concentration	Effects	Reference
HUVEC	1–100 µM	NADPH oxidase activity↓	[72]
HUVEC	10–100 µM	SOD1↑; GPx1↑; Nox4↓	[73]
EA.hy 926	100 µM	SOD1↑; SOD2↑; SOD3↑; GPx1↑ catalase↑	[71]
RAS	1–100 µM	GPx1↑ catalase↑	[74]
HCAEC	1–10 µM	SOD2↑; SIRT1↑; GSH↑; mtROS↓	[75]
HCAEC	0.1–100 µM	Nrf2↑; NQO1↑; GCLC↑; HO-1↑	[16]

GCLC, γ-glutamylcysteine synthetase; GPx1, glutathione peroxidase 1; HCAEC, human coronary arterial endothelial cells; HO-1, heme oxygenase-1; HUVEC, human umbilical vein endothelial cells; NQO1, NAD(P)H:quinone–oxidoreductase 1; RAS, rat aortic segments; mtROS, mitochondrial reactive oxygen species.

**Table 3 ijms-20-02155-t003:** Resveratrol reduces endothelin-1 synthesis.

Cell Type	Effective Concentration	Effects	Reference
HUVEC	1–100 µM	ROS↓; p-Erk1/2↓; strain-induced ET-1↓	[72]
HUVEC	0.1 µM	ET-1↓; eNOS↑; VEGF↑	[79]
HUVEC	30 µM	ET-1↓; ECE-1↓	[80]
HASMC	100 µM	H_2_O_2_-induced ET-1↓;	[81]
RASMC	10–100 µM	AngII-induced ET-1↓; proliferation↓	[82]

AngII, angiotensin II; ECE-1, endothelin-converting enzyme-1; HASMC, human aortic smooth muscle cells; HUVEC, human umbilical vein endothelial cells; RASMC, rat aortic smooth muscle cells; ROS, reactive oxygen species.

**Table 4 ijms-20-02155-t004:** Effects of resveratrol in vascular smooth muscle cells (VSMC).

Cell Type	Effective Concentration	Effects	Reference
VSMC	50–100 µM	Serum- and PDGF-induced proliferation↓	[94]
RASMC	0.1–1 µM	AGEs-stimulated proliferation↓	[95]
RASMC	25–50 µM	AngII-induced proliferation↓; p-Akt↓	[96]
HASMC	1–100 µM	Proliferation↓; p53↑; cell cycle arrest without apoptosis at 6.25–12.5 µM; apoptosis at 25 µM	[97]
RASMC	50–100 µM	Serum-induced proliferation↓; cell cycle arrest	[98]
RASMC	10–100 µM	AngII-induced proliferation↓; ET-1↓	[82]
BASMC	10–100 µM	Serum-induced proliferation↓; cell cycle arrest	[99]
HASMC	20–100 µM	TNF-α-induced proliferation↓; cell cycle arrest	[100]
HASMC	10–50 µM	Proliferation↓; p53↑; HSP27↑	[101]
RFSMC	25–50 µM	oxLDL-induced proliferation↓; PI3K/Akt/mTOR/p70S6K↓	[102]
HASMC	5–20 µM	PI3K activity↓; proliferation↓	[103]
RASMC	3–100 µM	Nrf2↑, HO-1↑; cyclin D↓, proliferation↓	[84]
HVSMC	3–100 µM	Differentiation of de-differentiated VSMC to the contractile phenotype	[86]
RASMC	50 µM	TGF-β-stimulated SMC de-differentiation↓; p-Akt↓; p-mTOR↓; KLF5↓	[93]

AGEs, advanced glycation end-products; AngII, angiotensin II; BASMC, bovine aortic smooth muscle cells; HASMC, human aortic smooth muscle cells; HVSMC, human vascular smooth muscle cells; RASMC, rat aortic smooth muscle cells; RFSMC, rabbit femoral smooth muscle cells.

**Table 5 ijms-20-02155-t005:** Resveratrol reduces blood pressure in animal models.

Model	Resveratrol dose	Effects	Reference
SHR	5 mg/kg (50 mg/L in drinking water) for 10 weeks	BP↓; ROS↓; 3-NT↓; EF↑; eNOS↑; eNOS uncoupling↓	[123]
SHR	146 mg/kg (4 g/kg mixed in chow) for 5 weeks	BP↓; FMD↑; p-AMPK↑; p-eNOS↑; 4-HNE↓	[124]
AngII-infused mouse	320 mg/kg (4 g/kg mixed in chow) for 2 weeks	BP↓; FMD↑; p-AMPK↑; p-eNOS↑; 4-HNE↓	[124]
Partially nephrectomized rats	50 mg/kg/day mixed in diet for 4 weeks	BP↓; NO↑; ET-1↓; AngII↓	[125]
Two-kidney, one-clip rats	10 mg/kg i.p. for 6 weeks	BP↓; EF↑; plasma TAC ↑; NO↑; tissue SOD↑, catalase↑, GSH↑, MDA↓ cardiac hypertrophy↓	[126]
DOCA salt	1 mg/kg by gavage for 32 days	BP↓; EF↑	[127]
Zucker rats	10 mg/kg by gavage for 8 weeks	BP↓; eNOS↑; TG↓; TC↓; insulin↓; leptin↓	[132]
HFD-fed female rats	20 mg/kg/day mixed with diet for 8 weeks	BP↓; EF↑	[129]
Fructose-fed rats	10 mg/kg by gavage for 45 days	BP↓; cardiac hypertrophy↓ eNOS↑; TBARS↓	[130]
HFCS-induced MetS in rats	5 mg/day (50 mg/L in drinking water) for 10 weeks	BP↓; TG↓; EF↑; p-eNOS↑; ROS↓	[131]
Ovariectomized rats	5 mg/kg by gavage for 3 weeks	BP↓; EF↑	[133]
Obese rats programmed by early weaning	30 mg/kg/day for 30 days	BP↓; TG↓; LDL↓; plasma MDA↓, SOD↑, catalase↑	[128]

3-NT, 3-nitrotyrosine; 4-HNE, 4-hydroxy-2-nonenal; BP, blood pressure; EF, endothelial function; HFCS, high-fructose corn syrup; HFD, high-fat diet; LDL, low-density lipoprotein; MDA, malondialdehyde; MetS, metabolic syndrome; SHR, spontaneously hypertensive rats; TAC, total antioxidant capacity; TBARS, thiobarbituric acid reactive substances; TC, total cholesterol; TG, triglycerides.

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
