# Peer review of "Resveratrol and Vascular Function"

_ijms, 2019, doi:10.3390/ijms20092155_

Reviewer 1 Report

In this review, the author Huige Li et al., have explained the role and molecular mechanism of Resveratrol and its effect on vascular function in a detailed manner. The manuscript is well written and well-structured article. It will be more interesting if the author can address a few minor remarks regarding the clinical perspectives and future scope of Revesterol.

1.       The author has explained in detailed how resveratrol will work in different cell type, also outlined different animal studies in a table. It would be interesting if the author can add a paragraph or table regarding current clinical trials or clinical research regarding Resveratrol in patients.

2.       The amount of resveratrol ingested from dietary sources, such as red wine and juices, results in plasma levels that are either not detectable or several orders of magnitude below the micromolar concentrations that are used in experimentation in vitro ≈32 nmol/L to 100 μmol/L.  Does Author think resveratrol will have a similar effect in human at low concentration?

3.       The Author needs to highlight the effect of resveratrol on glucose metabolism and insulin sensitivity.

Author Response

Response to Reviewer 1

Comments and Suggestions for Authors

In this review, the author Huige Li et al., have explained the role and molecular mechanism of Resveratrol and its effect on vascular function in a detailed manner. The manuscript is well written and well-structured article. It will be more interesting if the author can address a few minor remarks regarding the clinical perspectives and future scope of Resveratrol.

1.       The author has explained in detailed how resveratrol will work in different cell type, also outlined different animal studies in a table. It would be interesting if the author can add a paragraph or table regarding current clinical trials or clinical research regarding Resveratrol in patients.

Response:

A new section 7 (Effects of resveratrol on vascular function in human subjects) and a new table 6 have been added to the revised version.

2.       The amount of resveratrol ingested from dietary sources, such as red wine and juices, results in plasma levels that are either not detectable or several orders of magnitude below the micromolar concentrations that are used in experimentation in vitro ≈32 nmol/L to 100 μmol/L.  Does Author think resveratrol will have a similar effect in human at low concentration?

Response:

A new section 8 (Resveratrol bioavailability) has been added to address the reviewer’s concern. 

3.       The Author needs to highlight the effect of resveratrol on glucose metabolism and insulin sensitivity.

Response:

We agree with the reviewer that the effects of resveratrol on glucose metabolism and insulin sensitivity are an important issue. But we would like to ask the reviewer to consider that the focus of the present article is on vascular function. Glucose metabolism is not directly related. In addition, another article in the same IJMS special issue has systematically review this topic (see: Ref. [134] Hou et al. The Effects of Resveratrol in the Treatment of Metabolic Syndrome. Int J Mol Sci 2019, 20, (3) 535).

Nevertheless, we have included the effects of resveratrol on vascular function in clinical studies performed in patients with metabolic syndrome or diabetes (new Table 6).

Reviewer 2 Report

For the Authors:

Abstract:

The Abstract is a listing of potential effects of Resveratrol and the reader don’t get the information to decide if the article is of interest. The motivation for the review is missing.

Furthermore, there is well written Review available since 2017/2018 (https://www.nature.com/articles/s41698-017-0038-6). According to this review, very relevant limitations of the effect and side effects of Resveratrol haven’t been mentioned in this abstract.  

Until line 55, the whole section is listing of different potential target molecules of R.

I would have preferred a more general introduction leading to a more focused assessment of relevant target molecules, which will be discussed later on.

Line 59: Reference required. As this article is a review focusing on R. and endothelium, this section is too superficial (57-65).

Figure 1 and 2 look very similar, although the content is different; I would suggest to change the layout to avoid misunderstanding.

Table 3 is very informative and clearly structured.

199: remove one “in”

223: This is really relevant; the low bioavailability should be discussed extensively from a clinical point view regarding the applicability of R. as supplement or even drug.

259: This statement is not completely right; as leukocytes a very relevant for atherosclerosis and intimal injury, their relevance for intimal function is very relevant. Your statement, that they are not residential cells could be misleading in this context.

265 ff: I would recommend to add clinical implications of these findings: Can you report any studies wo reported a positive effect of R. and major cardiac events such as stroke, MI, acute limb ischemia and so on?

In general: I would recommend more clinical implications of all cellular findings as your manuscript name is: R. and Vascular function. A link to the vascular function and its pathophysiology should be discussed more intensively.

Conclusion:

Nothing to add.

Author Response

Response to Reviewer 1

Comments and Suggestions for Authors

In this review, the author Huige Li et al., have explained the role and molecular mechanism of Resveratrol and its effect on vascular function in a detailed manner. The manuscript is well written and well-structured article. It will be more interesting if the author can address a few minor remarks regarding the clinical perspectives and future scope of Resveratrol.

1.       The author has explained in detailed how resveratrol will work in different cell type, also outlined different animal studies in a table. It would be interesting if the author can add a paragraph or table regarding current clinical trials or clinical research regarding Resveratrol in patients.

Response:

A new section 7 (Effects of resveratrol on vascular function in human subjects) and a new table 6 have been added to the revised version.

2.       The amount of resveratrol ingested from dietary sources, such as red wine and juices, results in plasma levels that are either not detectable or several orders of magnitude below the micromolar concentrations that are used in experimentation in vitro ≈32 nmol/L to 100 μmol/L.  Does Author think resveratrol will have a similar effect in human at low concentration?

Response:

A new section 8 (Resveratrol bioavailability) has been added to address the reviewer’s concern. 

3.       The Author needs to highlight the effect of resveratrol on glucose metabolism and insulin sensitivity.

Response:

We agree with the reviewer that the effects of resveratrol on glucose metabolism and insulin sensitivity are an important issue. But we would like to ask the reviewer to consider that the focus of the present article is on vascular function. Glucose metabolism is not directly related. In addition, another article in the same IJMS special issue has systematically review this topic (see: Ref. [134] Hou et al. The Effects of Resveratrol in the Treatment of Metabolic Syndrome. Int J Mol Sci 2019, 20, (3) 535).

Nevertheless, we have included the effects of resveratrol on vascular function in clinical studies performed in patients with metabolic syndrome or diabetes (new Table 6).

Round  2

Reviewer 2 Report

All my comments have been addressed appropriately.

I have Nothing to add, congratulations.